# Categorizing Acute Respiratory Distress Syndrome with Different Severities by Oxygen Saturation Index

**DOI:** 10.3390/diagnostics14010037

**Published:** 2023-12-24

**Authors:** Shin-Hwar Wu, Chew-Teng Kor, Shu-Hua Chi, Chun-Yu Li

**Affiliations:** 1Division of Critical Care Internal Medicine, Department of Emergency Medicine and Critical Care, Changhua Christian Hospital, Changhua 50006, Taiwan; 2Big Data Center, Changhua Christian Hospital, Changhua 50006, Taiwan; 179297@cch.org.tw; 3Graduate Institute of Statistics and Information Science, National Changhua University of Education, Changhua 50006, Taiwan; 4Section of Respiratory Therapy, Department of Emergency Medicine and Critical Care, Changhua Christian Hospital, Changhua 50006, Taiwan; 31940@cch.org.tw (S.-H.C.); 181285@cch.org.tw (C.-Y.L.)

**Keywords:** acute respiratory distress syndrome, classification, mortality, oxygen saturation index

## Abstract

The oxygen saturation index (OSI), defined by F_I_O_2_/S_p_O_2_ multiplied by the mean airway pressure, has been reported to exceed the Berlin definition in predicting the mortality of acute respiratory distress syndrome (ARDS). The OSI has served as an alternative to the Berlin definition in categorizing pediatric ARDS. However, the use of the OSI for the stratification of adult ARDS has not been reported. A total of 379 invasively ventilated adult ARDS patients were retrospectively studied. The ARDS patients were classified into three groups by their incidence rate of mortality: mild (OSI < 14.69), moderate (14.69 < OSI < 23.08) and severe (OSI > 23.08). OSI-based categorization was highly correlated with the Berlin definition by a Kendall’s tau of 0.578 (*p* < 0.001). The Kaplan–Meier curves of the three OSI-based groups were significantly different (*p* < 0.001). By the Berlin definition, the hazard ratio for 28-day mortality was 0.58 (0.33–1.05) and 0.95 (0.55–1.67) for the moderate and severe groups, respectively (compared to the mild group). In contrast, the corresponding hazard ratio was 1.01 (0.69–1.47) and 2.39 (1.71–3.35) for the moderate and severe groups defined by the OSI. By multivariate analysis, OSI-based severe ARDS was independently associated with 28-D or 90-D mortality. In conclusion, we report the first OSI-based stratification for adult ARDS and find that it serves well as an alternative to the Berlin definition.

## 1. Introduction

The Berlin definition is currently the most widely accepted standard for the diagnosis and classification of acute respiratory distress syndrome (ARDS). However, there are some shortcomings to this definition.

First, its prognostic prediction ability is far from satisfactory. The area under the receiver operating characteristic (AUROC) of mortality prediction was only around 0.57 [1,2], which was just slightly larger than that under chance. The severity classification of the Berlin definition is based solely on the initial P_a_O_2_/F_I_O_2_, which has been found to be poorly associated with mortality in patients with ARDS in many studies [3,4,5]. Several important factors with potential prognostic implications are neglected in the Berlin definition. For one thing, high inflation airway pressure can increase mechanical stress on the lung and the chances of ventilator-induced lung injury [6,7,8]. Lower airway pressure has been shown to be associated with survival benefits of ventilated patients with ARDS [9,10]. By incorporating mean airway pressure (MAP) into P_a_O_2_/F_I_O_2_, the oxygenation index (OI) is calculated using the equation
$$OI=\frac{F_{I}O_{2}\times MAP\times 100}{P_{a}O_{2}}$$

The OI has been found to be better than P_a_O_2_/F_I_O_2_ in predicting the mortality of ARDS patients [11,12].

Another drawback of the Berlin definition is that its application is confined to settings when P_a_O_2_ is available. P_a_O_2_ can only be obtained sporadically by arterial puncture, which is painful and potentially harmful to patients. The absence of arterial blood gas data during crucial hypoxemic episodes may lead to a misclassification of the severity of ARDS in a patient. In areas where arterial blood analysis is unavailable, the prevalence of ARDS is inevitably under-reported. On the contrary, S_a_O_2_ can be continuously monitored by noninvasive pulse oximeters, which are ubiquitous in most ICUs. The S_a_O_2_/F_I_O_2_ and P_a_O_2_/F_I_O_2_ ratios are highly correlated [13] and provide similar prognostic information [14]. They also have similar cut-off points to identify mild and moderate ARDS [15,16]. The S_a_O_2_/F_I_O_2_ ratio has served as an alternative to the P_a_O_2_/F_I_O_2_ ratio in defining pediatric ARDS [17] and adult ARDS in resource-limited settings [18,19].

The oxygen saturation index (OSI) is generated by adding MAP to the S_a_O_2_/F_I_O_2_ ratio and is calculated using the equation
$$OSI=\frac{F_{I}O_{2}\times MAP\times 100}{S_{p}O_{2}}$$

The OSI can be non-invasively obtained and performs better than S_a_O_2_/F_I_O_2_, P_a_O_2_/F_I_O_2_ or the OI in predicting the mortality rate of ARDS [20,21]. The OSI can also be helpful in stratifying mortality risk for ARDS patients. The AUROCs to diagnose the P_a_O_2_/F_I_O_2_ ratio of less than 100, 200 and 300 with the OSI were 0.922, 0.869 and 0.787, respectively [22]. Using the OSI instead of P_a_O_2_/F_I_O_2_ to define and categorize ARDS can bypass the aforementioned drawbacks of the Berlin definition. The OSI has already been used to define and categorize pediatric ARDS with different severities [17]. For adults, the role of the OSI in the stratification of risk in patients with ARDS has not been studied enough. In this study, we tried to evaluate using the OSI to categorize adult ARDS patients with different severities.

## 2. Materials and Methods

### 2.1. Patient Enrollment

We retrospectively collected the data of invasively ventilated patients with ARDS admitted to Changhua Christian Hospital, a medical center with a total of 130 ICU beds distributed in 5 separate wards, between January 2012 and November 2018. These patients were identified by screening discharge diagnoses of ARDS and acute respiratory failure in electronic archives. Each diagnosis of ARDS was defined by the Berlin definition [1] and was reconfirmed by a pulmonologist. Exclusion criteria included age less than 20 or over 90 years old, body weight less than 40 or over 100 Kg, a total duration of invasive ventilation less than 48 h, absence of retrievable data of arterial blood gas or MAP in the first 3 days of ARDS diagnosis, using airway pressure release ventilation or high-frequency oscillation ventilation or extracorporeal membrane oxygenation during the ARDS period, co-morbidities of metastatic malignancy, congestive heart failure (left ventricular ejection fraction less than 35%) or ventilator dependence (invasive ventilation for 21 days or more before the onset of ARDS), having been transferred to other hospital or discharged against medical advice (without traceable clinical outcome), having been withdrawn from the life-support due to hospice, having been enrolled in other ARDS-related clinical studies and absence of need for lung-protective ventilation, which was defined by F_I_O_2_ ≥ 50% and PEEP > 5 cmH_2_O [23]. The patients’ data were traced until death or the 90th day after the diagnosis of ARDS. This study was approved by the institutional review board of Changhua Christian Hospital (approval number 191228).

### 2.2. Data Collection

Baseline variables when ARDS was diagnosed for the first time were collected. They include age, sex, body mass index (BMI), acute physiology and chronic health evaluation II (APACHE II) score, sequential organ failure assessment (SOFA) score, comorbidity, risk factors for ARDS and type of ICU admitted. Whether patients received sedation, muscle relaxant, systemic steroid, vasopressor, hemodialysis, prone position or total parenteral nutrition during the ARDS period was recorded. If the patients were under pressure-targeted ventilation and plateau pressures were not measured directly, the peak airway pressure or the sum of PEEP and the set increment of inspiratory pressure were used to represent plateau pressure [23].

### 2.3. Derivation of OI, OSI and Other Indices

The highest MAP and lowest P_a_O_2_/F_I_O_2_ and S_a_O_2_/F_I_O_2_ in the initial 3 days after ARDS diagnosis were used to calculate OI and OSI. The equation for calculating OI and OSI was mentioned in the previous section. We also adopt the lowest P_a_O_2_/F_I_O_2_ and S_a_O_2_/F_I_O_2_ of the initial 3 days as the commonly used predictor of ARDS mortality. Categorization according to Berlin definition was based on the lowest P_a_O_2_/F_I_O_2_ of the initial 3 days.

### 2.4. Statistical Analysis

Scatter plots and Spearman’s rho correlation analysis were used to present the linear correlation of OI and OSI. Categorical and continuous variables were expressed as numbers (proportions), mean ± standard deviation (SD) and median and interquartile range (IQR), respectively. The discriminating abilities of OI, OSI, S_p_O_2_/F_I_O_2_, P_a_O_2_/F_I_O_2_, APACHE II score and Berlin definition regarding mortality were assessed using receiver operating characteristic (ROC) curves and the corresponding AUROCs. Furthermore, ROC was also used to evaluate the OI and OSI categories with respect to P_a_O_2_/F_I_O_2_ of less than 200 or 100, respectively. The incidence rate per 100 person–days was used to visualize the trend of the hazard ratio for death over continuous values of OSI and OI. OI and OSI values were classified into low, moderate and high groups based on similar magnitudes of hazard according to the incidence rate per 100 person-days. Kendall’s tau correlation was calculated to evaluate the correlation between the categorizations by P_a_O_2_/F_I_O_2_ ratio-based Berlin definition, OI category and OSI category. Kaplan–Meier curves of estimated 28-day and 90-day survival were plotted and differences between the three groups were compared using the log-rank test. Survival analyses were performed to assess the association of OI, OSI levels and groups with mortality, using the low group category as the reference. According to the OI and OSI groups, crude and multivariate Cox proportional hazard models were constructed to estimate the mortality risk during the follow-up period. Statistical analyses were performed using SAS, and a visualization plot was performed using the R software (version 4.1.0 accessed on 18 May 2021; The Comprehensive R Archive Network: http://cran.r-project.org). All two-sided *p*-values less than 0.05 were considered statistically significant.

## 3. Results

### 3.1. Clinical Characteristics of Patients with ARDS

A total of 786 patients were found with invasive ventilation for ARDS and acute respiratory failure. Four hundred and seven patients were excluded due to extreme body weight for 20 of them, absence of traceable clinical outcomes for 19 of them, having been invasively ventilated for less than 48 h for 175 of them, other terminal comorbidities for 102 of them, absence of arterial blood gas data in the initial 3 days for 3 of them, having been ventilated by special modes for 2 of them, having received extracorporeal membrane oxygenation during the ARDS period for 70 of them, absence of ventilator settings eligible for lung-protective ventilation for 12 of them and having been withdrawn from a life-sustaining machine for hospice for 4 of them. Therefore, 379 patients were analyzed. The clinical characteristics of these patients are summarized in Table 1.

### 3.2. Correlation between OI and OSI

The OI and OSI were correlated by OI = −4.449 + 1.406 OSI and Spearman’s rho of 0.844 (*p* < 0.001) (Appendix A).

### 3.3. Comparison of ROC Curves of Commonly Used Indices in the Prediction of Mortality

The ROC curves of six commonly used indices, i.e., OI, OSI, S_p_O_2_/F_I_O_2_, P_a_O_2_/F_I_O_2_, APACHE II score and the Berlin definition, for predicting the mortality of ARDS patients, are depicted in Figure 1. The AUROC of the OSI was 0.630 (95% CI: 0.57–0.69) in predicting mortality at 28 days. This value was higher than that of any other index. In terms of predicting 90-day mortality, the AUROC of the OSI was 0.621 (95% CI: 0.56–0.68). Again, this AUROC was the largest among the six commonly used indices.

### 3.4. Using the OI or OSI Category to Diagnose P_a_O_2_/F_I_O_2_ Less Than 200 or 100

The AUROC of the OI category for diagnosing P_a_O_2_/F_I_O_2_ of less than 200 and 100 was 0.79 (0.72–0.87) and 0.94 (0.92–0.97), respectively. The AUROCs of the OSI category in diagnosing P_a_O_2_/F_I_O_2_ of less than 200 and 100 were 0.76 (0.68–0.84) and 0.84 (0.79–0.88), respectively (Appendix A).

### 3.5. Using the OI and OSI Values to Categorize ARDS with Different Severities

The incidence rates of mortality for every 100-person-day were plotted against continuous OI and OSI values divided by each 10 percentiles, as seen in Figure 2. Based on incidence rates of mortality, the types of ARDS were categorized into three mutually exclusive groups: mild (OI < 15.91, or OSI < 14.69), moderate (15.91 < OI < 28.78 or 14.69 < OSI < 23.08) and severe (OI > 28.78, or OSI > 23.08).

### 3.6. Correlations between OI/OSI-Based and P_a_O_2_/F_I_O_2_-Based (Berlin Definition) Categorization

OI or OSI-based categorization was highly correlated with P_a_O_2_/F_I_O_2_ ratio categorization by a Kendall’s tau of 0.754 (*p* < 0.001) (for the OI) or 0.578 (*p* < 0.001) (for the OSI) (Figure 3).

### 3.7. Mortality of Various OI/OSI-Based Severity Categories

The Kaplan–Meier curves of ARDS patients with various OI or OSI-based severity categories were significantly different for 28-day or 90-day mortality. All of these log-rank *p*-values were less than 0.001 (Figure 4). The mortality rate for each OI- or OSI-based severity category is presented in Appendix A.

### 3.8. Univariate Analyses of Variables Associated with Mortality

By univariate analysis, the selected variables potentially associated with mortality at 28 days are presented in Table 2.

The OI and OSI were both associated with increased mortality. However, the traditional ICU severity index (APACHE II score, SOFA score) or lung injury score was not associated with mortality at 28 days. Severe ARDS classified by OI > 28.78 had a 28-D mortality hazard ratio (HR) of 2.37 (1.73–3.26) over the mild counterpart (OI < 15.91) (*p* < 0.01). Severe ARDS classified by OSI > 23.08 had a 28-D mortality HR of 2.14 (1.58–2.91) over the mild counterpart (OSI < 14.69) (*p* < 0.01). In contrast, the 28-D mortality HR of severe ARDS classified by the Berlin definition was 0.89 (0.53–1.47) over the mild counterpart, which was not significantly different (*p* = 0.64). The contributors to 90-day mortality are presented in Appendix A. The data are similar to those of mortality at 28 days.

### 3.9. Multivariate Cox Proportional Hazard Analyses of Variables Associated with Mortality

Multivariate Cox proportional hazard analyses found that the OI (or OSI) as a continuous value and respiratory system compliance (C_RS_) were independently associated with mortality at 28 days. When patients were divided into three groups of severity based on the OI (or OSI), the severe group (versus mild) and C_RS_ were independent factors associated with mortality at 28 days. The OI (or OSI) was also independently associated with mortality at 90 days (Table 3).

## 4. Discussion

In this study, we report the first OSI-based mortality risk stratification in adult ARDS patients. This novel categorization showed a good correlation the with P_a_O_2_/F_I_O_2_-based Berlin definition (Figure 3), but it did not require an invasive technique and could be conveniently obtained. Furthermore, this OSI-based categorization was found to be superior to the P_a_O_2_/F_I_O_2_-based Berlin definition in discriminating the risk of mortality at 28 or 90 days (Table 2, Appendix A). Therefore, we think that the OSI could be an alternative to the P_a_O_2_/F_I_O_2_-based Berlin definition in categorizing adult ARDS with different severities.

Our study also demonstrates that the OSI has the largest AUROC in discriminating either 28-day or 90-day mortality, better than the other five commonly used predictors, i.e., OI, S_p_O_2_/F_I_O_2_, P_a_O_2_/F_I_O_2_, APACHE II score and the Berlin definition in this study (Figure 1). This result was in congruence with several previous reports [20,21]. For the detection of hypoxemia, S_a_O_2_ (or S_p_O_2_) is less sensitive than P_a_O_2_ when S_a_O_2_ is above 97%, but S_p_O_2_ is a reliable predictor of P_a_O_2_ in most other clinical circumstances [24]. Unlike sporadically sampled P_a_O_2_, S_p_O_2_ is continuously monitored and less likely to miss any hypoxemic episode. Missing P_a_O_2_/F_I_O_2_ data during significant hypoxemia may lead to the underestimation of severity and inaccurate prognostic prediction. We think that this may be the main reason why S_p_O_2_-based OSI outperforms other P_a_O_2_-based indices in prognostic prediction. Our speculation gains support from a large retrospective study including more than 35,000 patients. This study found that substituting missing P_a_O_2_/F_I_O_2_ data in the Sequential Organ Failure Assessment score with S_p_O_2_/F_I_O_2_ has greater discrimination ability for mortality than the miss-as-normal technique. The difference was most prominent in a subgroup of patients without baseline P_a_O_2_/F_I_O_2_ data [25].

Incorporating airway pressure into the equation may be another explanation for the OSI’s superior prognostic prediction ability compared to S_p_O_2_/F_I_O_2_, P_a_O_2_/F_I_O_2_ or the Berlin definition. Airway pressure is associated with lung compliance and ventilator settings, which surely contribute to outcomes of ARDS, as we have demonstrated in our multivariate analysis (Table 3). Barotrauma is the first well-studied mechanism of ventilator-induced lung injury [26]. Animal and clinical studies have shown that high airway pressure can induce pulmonary alveolar damage [6,7,8]. On the contrary, lower airway pressure has been shown to have a survival benefit in several large clinical trials [9,10].

DesPres et al. found that, when the OSI of their ARDS patients was greater than 19, the hospital mortality risk was greater than 30% [20]. Another study reported an adjusted odds ratio of 5.22 for mortality when the OSI was greater than 12 [21]. However, the above information is too indistinct to be applied in stratifying clinical ARDS cases. Here, we suggest unequivocal cut-off points (14.69 and 23.08) for defining three categories of ARDS, which are in line with usual clinical practice and the Berlin definition. The cutoff points that we suggested reflect the specific population group we collected in this study. This needs further verification by more studies with larger case numbers.

The outcomes of our mild and moderate groups are not significantly different no matter whether they are defined by the OI, OSI or Berlin definition. The hazard ratios of 28-D mortality for the moderate group (vs. mild) are 1.11, 1.01 and 0.58 by the OI, OSI and Berlin definition, respectively (Table 2). Since the discriminative ability of the Berlin definition has been proved [1], the failure to differentiate the two may be due to sampling-related type II error rather than the test itself. The inadequate number of mild ARDS cases (by the Berlin definition) may contribute to this error (Table 1). The correlation between the mild OI/OSI and P_a_O_2_/F_I_O_2_ ratio is not as significant as in the severe cases (Figure 3) can also be explained by this underrepresentation of the mild group. That is the first limitation of this study.

The second limitation of this study is that the patients were all collected from one medical center. To make the result of this study more relevant to patients with ARDS from other areas, we need more research with broader demographics.

Another limitation of this study is that only intubated patients with ARDS were enrolled. Non-invasive ventilators or high flow nasal oxygen are now more frequently applied to ARDS patients with a condition not severe enough to be intubated. The definition of pediatric ARDS has been broadened to include non-intubated patients [17]. The newly released global definition of ARDS also includes non-intubated adult patients [19]. We hope that future OSI or OI studies can incorporate this cohort of patients. However, accurately measuring airway pressure for OSI/OI calculation in non-intubated ARDS patients is an obstacle to overcome.

ARDS patients with COVID-19 were not included in our study. We are not confident that the results obtained from this study are applicable to this specific group of patients.

## 5. Conclusions

We present the first OSI-based severity stratification for adult patients with ARDS and find that it serves well as an alternative to the Berlin definition. The cut-off values we proposed need verification in future studies.

## Figures and Tables

**Figure 1 diagnostics-14-00037-f001:**
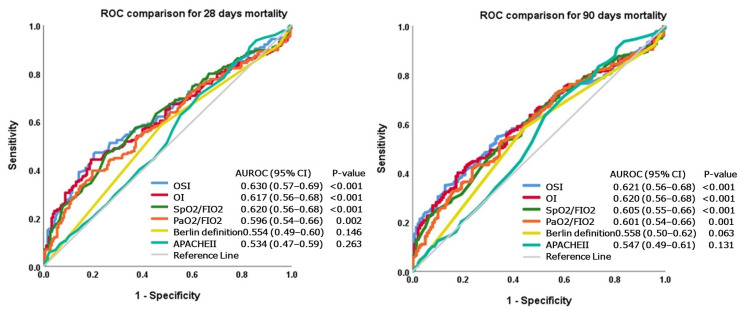
Comparison of ROC curves of six commonly used predictors of ARDS mortality. AUROCs of OSI were the highest for mortality at 28 days or 90 days.

**Figure 2 diagnostics-14-00037-f002:**
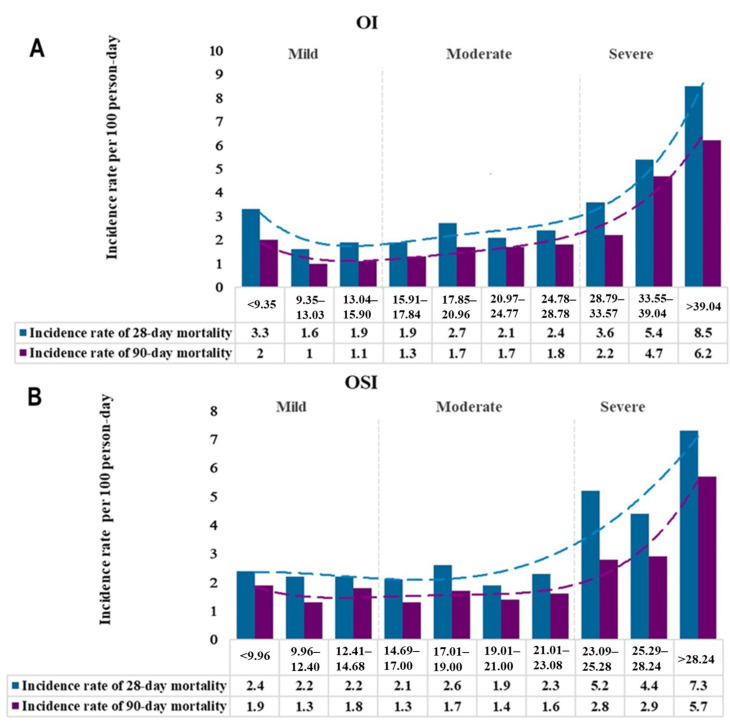
Incidence rate of mortality per 100-person-day stratified by the 10th percentiles of OI (**A**) and OSI (**B**). ARDS were classified into 3 groups: mild (OI < 15.91, or OSI < 14.69), moderate (15.91 < OI < 28.78 or 14.69 < OSI < 23.08) and severe (OI > 28.78, or OSI > 23.08).

**Figure 3 diagnostics-14-00037-f003:**
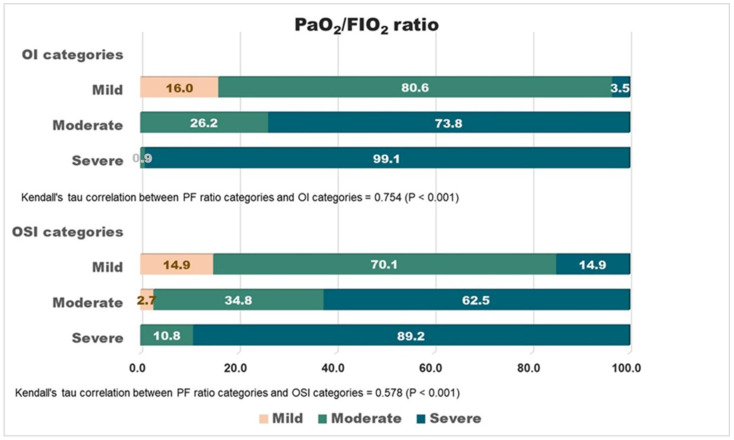
Correlation between OI/OSI-based and P_a_O_2_/F_I_O_2_-based categorizations by Kendall’s tau. Both were highly correlated by a Kendall’s tau of 0.754 (*p* < 0.001) (for OI) or 0.578 (*p* < 0.001) (for OSI).

**Figure 4 diagnostics-14-00037-f004:**
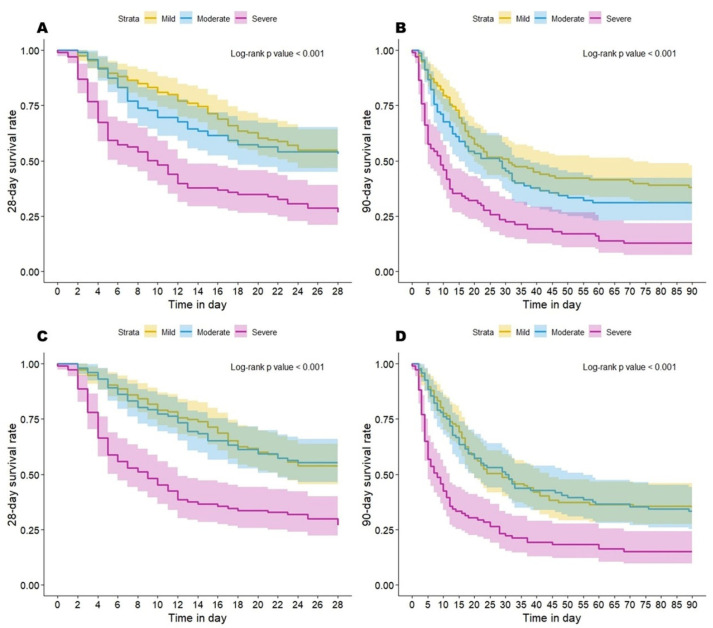
Kaplan–Meier curves of ARDS patients with various OI/OSI-based categories. ARDS patients categorized by OI for 28-day (**A**) or 90-day (**B**) survival. Kaplan–Meier curves of ARDS patients categorized by OSI for 28-day (**C**) or 90-day (**D**) survival.

**Table 1 diagnostics-14-00037-t001:** Clinical characteristics of the patients with ARDS (*n* = 379).

	Value
Age (year), mean ± SD	64 ± 16
Male, No. (%)	262 (69)
BMI, median (IQR), (Kg/m^2^)	23 (20–26)
APACHE II Score, median (IQR)	24 (18–30)
SOFA score, median (IQR)	7 (5–10)
Lung injury score, median (IQR)	11 (10–12)
Severity by Berlin definition	
Mild, No. (%)	23 (6)
Moderate, No. (%)	145 (38)
Severe, No. (%)	189 (50)
Missing, No. (%)	22 (6)
Comorbidity	
Chronic obstructive pulmonary disease, No. (%)	126 (33)
Diabetes mellitus, No. (%)	137 (36)
Hypertension, No. (%)	178 (47)
Chronic kidney disease, No. (%)	50 (13)
Heart failure, No. (%)	117 (31)
Cerebral vascular accident, No. (%)	90 (24)
Liver cirrhosis, No. (%)	44 (12)
Malignancy, No. (%)	88 (23)
Surgical ICU Admission, No. (%)	57 (15)
Treatment received during ARDS period	
Sedation, No. (%)	350 (92)
Muscle relaxant, No. (%)	356 (94)
Vasopressor, No. (%)	294 (78)
Total parenteral nutrition, No. (%)	75 (20)
Systemic steroid, No. (%)	320 (84)
Prone position, No. (%)	42 (11)
Hemodialysis, No. (%)	50 (13)
Continuous hemofiltration, No. (%)	126 (33)
Oxygenation index, median (IQR)	21 (15–31)
Oxygen saturation index, median (IQR)	19 (14–24)
V_T_/PBW ^1^ (mL/Kg), median (IQR)	9 (8–10)
P_a_O_2_/F_I_O_2_ ratio, median (IQR)	96 (71–133)
C_RS_ ^2^ (mL/cmH_2_O), median (IQR)	26 (22–31)
Plateau Pressure ^3^ (cmH_2_O), median (IQR)	32 (30–35)
PEEP (cmH_2_O), median (IQR)	10 (10–12)
Driving Pressure ^3^ (cmH_2_O), median (IQR)	21 (19–24)
28-day Mortality, No. (%)	186 (49)
90-day Mortality, No. (%)	233 (61)
Ventilator-free days, day 1–28 ^4^, median (IQR)	16 (6–22)

^1^ V_T_/PBW: tidal volume/predicted body weight. ^2^ C_RS_: respiratory system compliance. ^3^ Putative numbers, subject to over-estimation. See Section 2 or details. ^4^ In patients surviving by day 28.

**Table 2 diagnostics-14-00037-t002:** Univariate analysis of selected variables potentially associated with 28-day mortality.

	Survival	Death	HR (95% CI)	*p*-Value
Age (year), mean ± SD	62.4 ± 16.6	65.3 ± 15	1.00 (0.99–1.01)	0.40
Male, No. (%)	126 (65)	136 (73)	1.37 (0.99–1.90)	0.06
BMI, median (IQR), (Kg/m^2^)	23 (21–27)	23 (20–26)	0.97 (0.94–1.00)	0.07
APACHE II Score, median (IQR)	24 (18–30)	24 (19–29)	1.00 (0.99–1.02)	0.57
SOFA score, median (IQR)	7 (5–10)	7 (5–10)	1.02 (0.98–1.06)	0.29
Lung injury score, median (IQR)	11 (10–12)	11 (10–12)	1.01 (0.94–1.09)	0.82
Severity by Berlin definition				
Mild, No. (%)	9 (5)	14 (8)	1	1
Moderate, No. (%)	87 (48)	58 (33)	0.58 (0.33–1.05)	0.07
Severe, No. (%)	87 (48)	102 (59)	0.95 (0.55–1.67)	0.87
OI, median (IQR)	18 (13–27)	24 (16–35)	1.03(1.02–1.05)	<0.01 *
Mild, No. (%)	87 (47)	57 (33)	1	
Moderate, No. (%)	62 (34)	45 (26)	1.11 (0.75–1.64)	0.60
Severe, No. (%)	35 (19)	72 (41)	2.45 (1.73–3.47)	<0.01 *
OSI, median (IQR)	18 (13–22)	21 (14–26)	1.05 (1.03–1.07)	<0.01 *
Mild, No. (%)	90 (47)	63 (34)	1	
Moderate, No. (%)	67 (35)	47 (25)	1.01(0.69–1.47)	0.96
Severe, No. (%)	36 (19)	76 (41)	2.39 (1.71–3.35)	<0.01 *
Comorbidity				
Hypertension, No. (%)	98 (51)	80 (43)	0.74 (0.55–0.98)	0.04 *
Liver cirrhosis, No. (%)	15 (8)	29 (16)	1.61 (1.08–2.39)	0.02 *
Malignancy, No. (%)	32 (17)	56 (30)	1.54 (1.13–2.11)	0.01 *
Treatment received				
Vasopressor, No. (%)	128 (66)	166 (89)	3.12 (1.96–4.97)	<0.01 *
Hemodialysis, No. (%)	30 (16)	20 (11)	0.61 (0.38–0.97)	0.04 *
Continuous hemofiltration, No. (%)	42 (22)	84 (45)	1.99 (1.49–2.66)	<0.01 *
V_T_/PBW ^1^ (ml/Kg)	9 (8–10)	9 (8–10)	0.86 (0.8–0.94)	<0.01 *
C_RS_ ^2^ (ml/cmH_2_O)	28 (25–33)	24 (20–28)	0.94 (0.92–0.96)	<0.01 *
PEEP (cmH_2_O)	10 (9–12)	11 (10–13)	1.13 (1.06–1.22)	<0.01 *
Driving pressure ^3^ (cmH_2_O)	20 (18–23)	22 (20–25)	1.07 (1.03–1.10)	<0.01 *

^1^ V_T_/PBW: Tidal volume/predicted body weight. ^2^ C_RS_: Compliance of respiratory system. ^3^ Putative numbers, subject to over-estimation. See Section 2 for details. * *p* < 0.05.

**Table 3 diagnostics-14-00037-t003:** Multivariate Cox proportional hazard analysis of factors associated with mortality.

	28-Day Mortality ^1^				90-Day Mortality ^2^			
	aHR ^3^ (95% CI)	*p*	aHR (95% CI)	*p*	aHR (95% CI)	*p*	aHR (95% CI)	*p*
OI or OSI as continuous values								
OI	1.03 (1.01–1.04)	<0.01	-	-	1.04 (1.02–1.06)	<0.001	-	-
OSI	-	-	1.03 (1.01–1.05)	<0.01	-	-	1.04 (1.02–1.06)	<0.01
C_RS_	0.93 (0.91–0.96)	<0.01	0.94 (0.92–0.97)	<0.01	0.93 (0.90–0.95)	<0.001	0.93 (0.91–0.95)	<0.01
V_T_/PBW	1.06 (0.95–1.17)	0.31	1.06 (0.96–1.17)	0.26	1.15 (1.04–1.27)	0.007	1.12 (1.01–1.23)	0.03
OI or OSI as 3 groups								
OI group								
Mild	1		-	-	1		-	-
Moderate	1.05 (0.71–1.57)	0.80	-	-	1.47 (0.95–2.30)	0.09	-	-
Severe	2.24 (1.54–3.26)	<0.01	-	-	3.03 (1.85–4.95)	<0.01	-	-
OSI group								
Mild	-	-	1		-	-	1	
Moderate	-	-	0.98 (0.67–1.44)	0.94	-	-	0.96 (0.66–1.41)	0.84
Severe	-	-	2.26 (1.58–3.24)	<0.01	-	-	2.15 (1.43–3.24)	<0.01
C_RS_	0.93 (0.91–0.96)	<0.01	0.93 (0.91–0.96)	<0.01	0.93 (0.90–0.95)	<0.01	0.93 (0.90–0.95)	<0.01
V_T_/PBW	1.07 (0.96–1.19)	0.22	1.06 (0.96–1.17)	0.25	1.16 (1.06–1.28)	0.01	1.15 (1.05–1.26)	<0.01

^1^ Model was adjusted for gender, BMI, hypertension, liver cirrhosis, malignancy, vasopressor, hemodialysis and continuous hemofiltration. ^2^ Model was adjusted for age, gender, BMI, P_a_O_2_/F_I_O_2_ ratio, hypertension, liver cirrhosis, malignancy, vasopressor, systemic steroid, prone position and continuous hemofiltration. ^3^ Adjusted hazard ratio.

## Data Availability

The data presented in this study are available upon reasonable request from the corresponding author.

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
