# Peer review of "Categorizing Acute Respiratory Distress Syndrome with Different Severities by Oxygen Saturation Index"

_diagnostics, 2023, doi:10.3390/diagnostics14010037_

Round 1

Reviewer 1 Report

Comments and Suggestions for Authors

1.  These authors have analyzed the utility of the oxygen saturation index in predicting outcomes in patients with acute respiratory distress syndrome.  This study included 379 adult patients on mechanical ventilation.  94% of the patients had either moderate or severe ARDS based on the Berlin definition.  The median tidal volume for ideal body weight was 9.  The median plateau pressure was 32, the median PEEP level was 10, and the median driving pressure was 21. The 28-day mortality was 49%.
2.  There was an excellent correlation between the oxygenation index and the oxygen saturation index.  Oxygen saturation index predicted mortality better than the other parameters frequently used in these patients.  This was true in multivariate Cox hazard analysis using the parameter as a continuous variable or as a categorical variable.
3.  This study demonstrates that the oxygen saturation index can predict mortality in patients with ARDS.  The oxygen saturation value is easily available in all patients throughout their management in an ICU.  Consequently, it likely provides a better index of ongoing gas exchange than periodic arterial blood gases.
4.  This is a well-done study with important outcomes.  The authors should include the equation for the oxygen saturation index in the abstract.

Reviewer 2 Report

Comments and Suggestions for Authors

I reviewed the manuscript diagnostics-2726272. The authors examined the role of OI and OSI scores in predicting ARDS severity and 28-, 90-day mortality as well. They demonstrated that OSI categorization (a score determined by a non-invasive technique) is superior to Berlin criterion, which is relied to P/F ratio measurement. The study is well-designed and generally well-written. The most important disadvantages of the study are the following:

1) The authors should explain why underweight and overweight patients were excluded from the study.

2) The authors report that 84% of the patients were treated with corticosteroids. They should define the specific substances and duration.

3) The majority of the patients received non-protective ventilation according to ARDS net studies (i.e. Vt/PBW = 9 ml/kg). The authors should explain why.

4) In Figure 2A the authors should explain the increased rate of 28-day mortality in mild ARDS group (grouping by OI score).

5) In Figure 3 it is demonstrated that P/F ratio is better correlated with severe OI/OSI scores. This should be addressed in the discussion section.

6) In Figure 4 and Table 2 is evident that there is an overlap of mortality between mild and moderate OI/OSI groups and as it is shown in table 3 the severe OI and OSI scores have a better discrimination capacity for 28-, 90-day mortality. This should be addressed in the discussion section.

7) Citation by DesPres et al. has not been found   
